# Remote Communications between Patients and General Practitioners: Do Patients Choose the Most Effective Communication Routes?

**DOI:** 10.3390/ijerph20247188

**Published:** 2023-12-16

**Authors:** Ido Morag, Efrat Kedmi-Shahar, Dana Arad

**Affiliations:** 1School of Industrial Engineering and Management, Shenkar College of Engineering and Design, Ramat-Gan 5252626, Israel; 2Ministry of Health—State of Patient Safety Division, Ministry of Health—State of Israel, 39 Yirmiyahu St., P.O. Box 1176, Jerusalem 9446724, Israel; efrat@usense-ux.com; 3Clalit Health Services, Innovation Division, 40 Toval St., Ramat Gan 5252247, Israel; danaarad@clalit.org.il

**Keywords:** remote communication, treatment quality evaluation criteria, criterion-based analysis, patient–general practitioner remote and conventional consultation

## Abstract

The use of remote communication between patients and general practitioners has greatly increased worldwide, especially following the COVID-19 outbreak. Yet, it is important to evaluate the impact of this shift on healthcare quality. This study aimed at evaluating remote healthcare quality by comparing four remote patient-to-physician communication modes used in Israel. The research methodology entailed criteria-based analysis conducted by healthcare quality experts and a subjective patient-perception questionnaire regarding the healthcare quality attributed to each mode and the extent to which each mode was used. Our findings indicate that the extent to which each mode is used was found to be inversely related to its rated quality. As such, the common assumption whereby patients tend to choose the mode of communication that will most likely ensure high service quality is refuted. Our findings also indicate that remote services often hinder the physician’s understanding of the patient’s clinical issues, as patients encounter difficulties in correctly articulating and conveying them; such services also hinder the patient’s understanding of the recommended course of treatment. These findings should be addressed by policymakers for improving remote communication services to ensure optimal healthcare service quality.

## 1. Introduction

The COVID-19 pandemic greatly accelerated the integration of remote services in the practices of healthcare providers worldwide—services that are often referred to collectively as eHealth, telehealth, or telemedicine [1]. The use of eHealth has increased 38-fold from the pre-COVID-19 pandemic baseline, while face-to-face visits have decreased significantly [2]. This meaningful change was driven by the need to provide patients with diagnostic and therapeutic services while adhering to social distancing regulations [3]. For example, in Canada, eHealth services accounted for 71.1% of all primary care appointments during the first four months of the pandemic in 2020, compared to 1.2% in the same period in 2019 [4], while in Ireland, face-to-face meetings decreased by 47% during COVID-19, while telemedicine meetings increased by 543% [5].

Among the remote medical services now available are asynchronous remote-request messages, communicated between patients and their general practitioners (GPs) through an information technology platform, as a means of eliciting remote health services and receiving personalized feedback. The rate of such remote messages in Canada has been estimated at 36% of all eHealth services [6].

Remote requests are generally regarded as complementary to face-to-face meetings with the GP, rather than as a direct substitute, depending on the patients’ preferences and medical needs [7]. The use of such services is conditioned by the doctor’s prior acquaintance with the patient and can be utilized for a wide range of needs, from renewing a prescription, discussing laboratory test results, and requesting a face-to-face visit, to seeking a referral to a specialist, and following-up on chronic health conditions [8,9]. Yet, they are inadequate for addressing certain clinical needs, such as when the patient requires immediate intervention, a physical examination, or measurements of various parameters. Hence, it is the responsibility of the GP to determine whether a specific case requires a different mode of interaction, such as a video or in-person consultation.

### 1.1. Benefits of Remote Patient–GP Communication Modes

As the use of remote services has expanded worldwide [9,10], thereby changing the face of social practices [11], their benefits as a complementary supplement to GP outpatient visits are being increasingly recognized [12]. Indeed, benefits can be seen for the patient, the GP, the health service organization (HSO) that operates these services, and the government. From the GP’s perspective, fewer face-to-face visits may enable them to focus more on critically ill patients, deal with urgent consultations, and reduce their personal exposure to viruses and infections. In other words, the introduction of remote communications can enhance the GP’s autonomy and job satisfaction, free up resources, and reduce mental and physical stress [13].

From the patient’s perspective, remote access to their GP could offer a more convenient means of communication, while improving ease of access and equality, especially in remote rural areas that lack sufficient human capital and resources [10,14,15]. Moreover, remote communication may also provide more flexible access times, as patients can contact their GPs even if they are not at home; support expeditious and effective communications about treatment continuity [12]; enhance the speed of treatment by enabling the continuous managing and controlling of the patient’s clinical conditions [12,16]; and reduce transportation time costs as well as downtime—thereby maintaining work productivity and other benefits [17]. In addition, assuming that GPs enable the remote observation of signs and symptoms, either through the use of questionnaires or self-operated monitoring devices (e.g., blood glucose and blood pressure), the remote services could help to promote the patients’ self-management of their conditions and their sense of independence [12,18]. This is especially applicable in relation to chronic disease management and preventative healthcare [8].

For the HSO, these services have been found to be cost-effective (with reduced medical, administrative, and infrastructure expenditure), while offering improved treatment efficacy [1]. Some studies go so far as to suggest that in some cases, remote clinical services are at least as effective as face-to-face consultations [19]. Finally, for government bodies, the development and implementation of remote health services are key to transforming public health [20]. In addition, as eHealth services become more prevalent, governmental bodies will be better able to cope with shortages in healthcare resources, and may even see a reduction in the number of patient referrals to hospitals and unplanned hospitalizations [20].

Yet, alongside these benefits, eHealth services may also pose risks in terms of the quality of patient care [21]. These risks pertain to diagnostic accuracy and to the optimal management of ongoing conditions due to partial, unreliable, or missing information that may be associated with remote consultations [19]. Patients may not always be able to communicate their symptoms coherently, and clinical decision making may be impaired due to contextual limitations, such as the difficulty to assess urgency or select the optimal drug due to an insufficient examination of the patient.

Given the potential benefits of remote healthcare services and the possibility of additional pandemics in the future, it is likely that remote health communications will continue to expand over the coming years. According to the World Health Organization, advanced technologies and Internet connectivity offer new methods for utilizing and improving eHealth services and for enhancing the quality of medical care supported by such services [22]. For example, their capacity and functionality have been enhanced by integrating advanced technologies, such as artificial intelligence (AI), data analytics solutions, and decision-support systems, into the design of these services [23]. Yet while technological platforms and changes in healthcare service content are rapidly evolving, the literature is lacking a universal format or standard for developing and structuring such services, with an emphasis on preventing related risks to healthcare quality as a result of the transition to remote healthcare services.

### 1.2. The Aim of the Study

In light of this literature review, the aim of this study was to compare the four remote communication modes applied in Israel, using a set of criteria that pertain to the perceived healthcare quality achieved through each mode [24]. Modes 1 and 2 are official online HSO services that only differ in the type of communication that they allow (selecting from built-in menu options vs. typing free text). With Mode 3, the patient calls the clinic and asks a member of staff to forward a written request to the GP through an internal digital system. This mode is less formal, having been grafted over time onto the existing system of communications between the clinic’s personnel and its GPs. (Since telephone calls are often used for conveying medical queries to physicians, this mode has been included in this study, mainly for comparative purposes.) Mode 4 concerns sending a free-style written message directly to the GP’s cellphone or email. This mode is the least official mode of remote communications, and is not monitored or controlled by the HSOs; yet, while its use is not recommended, it is not banned, and the protocol for handling information through this informal channel is at the sole discretion of the GP [24].

This novel study could contribute to the literature by providing recommendations regarding the optimal mode and format of patient–GP remote communications, to benefit all parties involved while ensuring optimal healthcare quality. Moreover, the insights gained from this study could shed light on a range of important aspects of remote patient–GP communications, particularly for older patients who are more likely to suffer from chronic diseases [25] and other populations with greater medical needs.

To understand the setting of this study, it is important to note that all citizens in Israel are expected to be a member of one of four HSOs. Patients may choose their GP, yet cannot transfer to a different GP during that quarter, during which time they may consult that GP unlimitedly (both in-person and remotely). At the end of a given quarter, the patient may choose to transfer to a different GP from the same HSO. In practice, the rate of transfer between GPs is extremely low [26]. In terms of costs, patients pay a fixed quarterly fee to the HSO to which they belong. For each patient–GP consultation, the HSO is reimbursed by the State. Finally, in Israel, all GPs are employed by HSOs.

It should be noted that this study only examines asynchronous text communications in which the GP responds to the patient’s requests; no real-time and/or two-directional audio/video communications were assessed. As such, this research focuses on just one facet of the ever-evolving pattern of remote healthcare services.

## 2. Methods

In 2000, as part of a global initiative to develop eHealth [26], the Israeli Ministry of Health asked its four HSOs to develop modes for enabling remote patient–GP services. Yet, a lack of clear government guidelines led each HSO to implement remote services in an ad hoc manner, based solely on the personal preferences and experience of their GPs. In 2022, given the increasing use of such services due to the COVID-19 pandemic, the Ministry of Health conducted evaluations of these modes and their ability to provide quality healthcare. The following four modes of conveying remote requests to the GP were identified:

Mode 1, structured—initiating online communications with the GP via the HSO’s website, via pre-defined options presented in a built-in menu.

Mode 2, unstructured—initiating online communications with the GP via the HSO’s website, using free texts as desired.

Mode 3—initiating online communications with the GP by making a telephone call to the clinic and conveying a message to the GP via a member of staff, who then submits the query to the GP as a written message via an internal digital system.

Mode 4—initiating online communications with the GP by sending a written text message directly to the GP’s mobile phone or email.

### 2.1. Research Design and Participants

To assess the four modes of remote request communications, a seven-participant committee was formed, including three human factors experts from the Patient Safety Division of the Israeli Ministry of Health (each with more than 20 years’ experience in patient safety research), and a representative from the Treatment Safety and Risk Management Divisions of each of the four HSOs. Their activity comprised three consecutive stages.

### 2.2. A Criterion-Based Analysis of the Perceived Ability of the Four Communication Modes to Provide Quality Healthcare

Stage 1: The committee members were asked to identify factors that could predict the ability of each remote communication mode to provide quality healthcare. This was conducted through task analysis, grounded in the System Engineering Initiative for Patient Safety Model of Work System and Patient Safety, which examines how healthcare information technology applications affect workflows or processes that are products of the healthcare system [27]. At this stage, each of the seven committee members conducted interviews with GPs and clinic office personnel and carried out direct observations of their behavior when dealing with remote consultations and requests. It is important to note that due to the difficulty in interpreting messages without knowledge of the given medical context, the specific communication messages between the GPs and their patients were not analyzed.

Stage 2: The committee members presented the factors that they had identified as being pertinent to remote communications. Of the nine criteria presented in total, the members reached an agreement that two of these factors do not contribute to a better understanding of the quality of the healthcare provided in all four modes (i.e., both factors referred to technical problems that are not relevant to each of the modes, such as Internet infrastructure). To ensure research validity, great efforts were made to formulate and articulate the remaining seven criteria, with an emphasis on clarity and unambiguity within each criterion, and on distinct differences between the criteria. The following are the seven criteria that were selected for assessing the four modes of remote communications in this research:

(A) Is the patient’s identity clearly specified?

(B) Does the submission process allow the use of clear and comprehensive specifications of the medical issue that require the GP’s attention?

(C) Is the patient’s query automatically documented in the patient’s medical records?

(D) Are the patient’s medical records accessible to the GP when processing the query?

(E) Is the GP’s answer automatically documented in the patient’s medical records?

(F) Is the maximum timeframe in which the GP must respond to a query specified in advance?

(G) Is the GP’s response clearly communicated to the patient?

Stage 3: The committee members were asked to examine the four communication modes, based on the seven selected criteria. For each mode, they individually evaluated whether the mode satisfied each criterion *fully*, *normally* (i.e., most of the time), *partially* (i.e., some of the time), *rarely*, or *not at all*. To do so, they were asked to apply their acquired knowledge following their observations and interviews; their familiarity with the HSO’s protocols; and after consulting with clinical personnel. To ensure between-participant assessment consistency, Fleiss’ kappa inter-rater reliability analysis was conducted [28].

Following their individual assessments, the committee members were asked to reach a consensus regarding the ranking order of each communication mode in terms of its likely capacity for supporting adequate healthcare quality.

### 2.3. Patients’ Perceptions of Healthcare Quality via the Four Communication Modes

In the following stage, patients were asked to complete an online survey to evaluate how they perceive the quality of healthcare that they receive via the communication mode that they most frequently use. This stage was conducted in accordance with the Declaration of Helsinki, and the protocol was approved by the Ethics Committee of Bar-Ilan University, Israel. The questionnaire was adapted from the 40-item questionnaire that was developed and tested by Baudier et al., with Cronbach’s α = 0.900 and composite reliability = 0.930 [29]. Nine relevant items were selected from this questionnaire and their wording was adjusted to fit the context of the present study. For example, *The service allows me to maintain a trusting relationship with my GP*, or *I find remote communication to be more efficient than attending the clinic*.

A total of 400 respondents—100 users for each communication mode—were asked to complete the online questionnaire, which was distributed via email and social media platforms using Google Forms (Appendix A). The users were randomly selected from a pool of patients who had submitted a query for their GP at least three times during the previous quarter and using the same particular mode (e.g., remote mode 2, 2, 2, remote mode 4, 4, 4). That is, each respondent completed the questionnaire in relation to one specific remote mode of communication. Prior to answering the survey, the respondents were asked to provide their informed consent to participating in this research study. The respondents were then asked to rate the nine items on a Likert-like scale from 1 (strongly disagree) to 5 (strongly agree), to convey the extent to which the respondent perceives the specific goal as being met (e.g., that their medical issue is fully understood) through their chosen remote healthcare mode. All respondents were at least 18 years of age, as, below this age, patients are legally considered minors and all contact with the medical staff must be made through their parents.

Statistical analysis was conducted using SPSS [30]. One-way ANOVA tests were conducted to examine differences between patients’ perceptions of the four communication modes; a-prior power analysis was also conducted using G*Power3 [31], while applying a one-way ANOVA test and an alpha level of 0.05.

The ratings within each communication mode were averaged over patients and items to generate a ranking order of confidence in the perceived ability of the given mode to deliver a range of positive outcomes that are associated with quality healthcare.

### 2.4. The Use of Each Mode in Practice

To analyze the extent of patients’ actual use of each of the four communication modes in Israel, stratified by age, data was extracted from the records of the four HSOs during 2016–2021. Although these data refer to a period prior to the two other analyses (as more recent data was not available), they are significant for the study’s insights.

## 3. Results

### 3.1. Four Communication Modes: The Performance Assessment by Criteria

The between-participant consistency levels of the seven assessed criteria (A to G) were measured via the Fleiss’ kappa reliability test: A = 0.564; B = 0.473; C = 0.514; D = 0.460; E = 0.451; F = 0.520; and G = 0.508. These results reflect the moderate strength of between-participant agreement, with an average of 50% agreement *beyond* the expected chance agreement (Appendix B depicts the participants’ ratings). Moreover, the performance assessments of each communication mode via the seven criteria (i.e., criterion-based analysis, Table 1) demonstrate that all four modes entail certain potential risks as to healthcare quality, with Mode 4 (sending a written message to the GP’s mobile phone or email) presenting the greatest risks.

Substantial differences in the capacity to satisfy the quality-related criteria were found between the first two (online) and the last two modes. For Mode 1 and Mode 2, the criteria were *fully* met six and five times, respectively. On the other hand, for Mode 3, the criteria were only *normally* met once, and were *partially* met five times; Mode 4 did not meet any criterion *normally*, and *rarely* met most criteria. The performance assessment of each communication mode by criteria is presented in Table 1.

### 3.2. Patients’ Confidence in the Efficacy of the Four Communication Modes

Similar to the criterion-based analysis, only partial confidence was conveyed by the patients in their chosen communication mode’s ability to meet all nine quality outcomes. The average ratings on the five-point scale ranged between three and four across all modes and criteria (Table 2). This analysis is based on 176 completed questionnaires (41, 47, 36, and 52 completed questionnaires for Modes 1, 2, 3, and 4, respectively), representing an overall response rate of 44%. With an alpha level of 0.05, this analysis yielded the power of 89.6%, which is considered very high. (See Appendix C for the raw data that was used to comprise Table 2.)

Significant differences were found between the four modes (one-way ANOVA test, F_(3172)_ = 4.8, *p* = 0.003), with Mode 4 being the most problematic. Interestingly, item 3 (*I fully understand the medical instructions given by the GP through the remote service*), which produced the weakest ratings across all four modes, related to the expectation that patients will understand the instructions that they received from their GP. It could be that, in addition to patients not always being able to adequately formulate their health issues in writing, they may also encounter trouble interpreting their GPs’ instructions. Yet despite these difficulties, it appears that patients do perceive the healthcare quality that they receive via their chosen mode of communication as providing a reasonably adequate and efficient alternative to visiting to the clinic in person (see items 5 and 9 in Table 2).

### 3.3. The Interrelationships between the Criterion-Based Analysis, Patients’ Ratings, and the Extent of Use

When examining the healthcare quality, the criterion-based assessment in this study was based on seven features intrinsic to the four communication modes; the patient-perception assessments, on the other hand, were based on nine aspects that convey patients’ confidence in their chosen mode of communication for providing them with quality healthcare. Despite these differences, and as seen in Table 3, both assessments led to the exact same ranking order of the four remote communication modes: Mode 1 received the highest rating (i.e., the structured online messaging option), while Mode 4 received the lowest rating (i.e., sending an unstructured message to the GP’s personal phone or email). These findings suggest that the assessments are fairly robust, and could reasonably be regarded as proxy measures of the quality of care associated with these modes of communication.

However, both assessments were inversely related to the ranking of the extent of use (Table 3). This suggests that the frequency of use is not directly driven by quality expectations. Indeed, as our analysis shows, the most commonly used mode of communication, Mode 4, is that which generates the least confidence and is judged to be the one that provides the lowest quality of care.

### 3.4. The Relationship between Age and the Usage of the Remote Request Modes

Finally, we also assessed the extent of utilization of each mode by patient age, based on data extracted from the HSOs’ databases for 2016–2021 (Table 4). The findings show that patients under the age of 30 years were most likely to use Mode 4, i.e., send a free-style message directly to the GP’s cellphone or email [Chi-square (3) = 4562, *p* < 0.001], whereas users aged 45 years and above were most likely to use Mode 3, i.e., contact the clinic personnel to convey their communication to their GP [Chi-square (3) = 5577, *p* < 0.001].

## 4. Discussion

The aim of this study was to examine the four remote modes of communication between patients and their GPs in Israel, with an emphasis on those features of communication that might impact the quality of healthcare received via these modes.

The most prominent findings of this study were that the two online modes (Modes 1 and 2) were perceived as enabling better healthcare quality than the two other modes (Modes 3 and 4). This was seen in both the criteria-based analysis and in the ratings submitted by the respondents. In Modes 1 and 2, the method for conveying the medical query to the GP was structured or semi-structured, respectively. It seems that this ensures that the medical issue at hand is conveyed in a more coherent manner (i.e., correctly, accurately, and in full), and in turn elicits clearer guidance regarding the GP’s response and medical advice. With Modes 3 and 4, there is neither a structured form of submitting a medical query, nor a built-in mechanism to ensure that the GP follows up on the issue and provides the patient with clear instructions and advice. In other words, as the requests are not automatically integrated into the medical systems in Modes 3 and 4, the patient must rely on the GP’s availability, commitment, and prioritization, which are not always in line with the patient’s needs (especially when requests are made outside office hours). Most surprising is the finding whereby patients choose the most *convenient* remote communication mode, rather than the mode of remote communications that is most likely to maximize the quality of their received healthcare, as could be assumed [32,33]. However, these findings are consistent with more general ones that demonstrate that the usability of a service (e.g., ease of use and intuitiveness) outweighs other criteria in user preferences (e.g., the efficiency of the service or its expected benefit) [34,35]. It seems that patients are willing to forgo a certain degree of satisfaction with their received healthcare quality for the sake of convenience—presumably because of the value that they place on forgoing an actual visit to the clinic and on not having to deal with the more complex interfaces of the HSO website communication portals.

In regard to the relationship between age and the usage of the remote request modes, a possible explanation is that younger people prefer intuitive and familiar communication modes (i.e., free and informal messaging through personal appliances), perhaps driven by their experiences on social networks [35], while older users, who are known to experience difficulties operating digital systems [36] and to be more concerned about losing interpersonal contact [37,38], seem to prefer the intervention of the clinic’s personnel in order to communicate with their doctors.

### 4.1. The Benefits of Remote HSO-Structured Communications

Our primary conclusion is that the most adequate mode of remote communication is that which is based on a structured route via the HSO’s website (i.e., Mode 1). First, it fully met six of the seven criteria that are perceived to be related to healthcare quality, outperforming the other three modes. Mode 1 also elicited the greatest degree of confidence among patients that the communication will lead to constructive outcomes. In addition, its structured application can be modified and adapted to meet specific aims in the future, and with greater ease than the other modes (e.g., modifying definitions in the menus and workflows, while updating empirical research findings and outcomes regarding important health issues). Finally, this mode utilizes a service that is integrated in the HSO systems, thereby encouraging a more holistic approach to healthcare, while enabling the creation of patient portfolios. Such portfolios could help to promote a wide range of functions and services that are relevant to the patient, thereby improving healthcare quality and standards. Such services could, for example, include pregnancy monitoring, parental guidance, and diabetes workshops, as well as the conveying of up-to-date information on how to deal with new pandemics.

From the patients’ perspective, contacting their GP in a structured manner via their HSO website might also expose them to the full range of options that are available to them in relation to their medical management, thereby increasing patient awareness and again improving service quality. From the GP’s perspective, communicating through this mode enables medical decision making that is based on the patient’s evolving medical history. Finally, from a broader healthcare perspective, additional professionals (such as dietitians or physiotherapists) could have access to important medical data, enabling them to provide more tailored treatment to the individual patients.

### 4.2. Recommended Improvements in the Remote HSO-Structured Communications

Given the potential benefits of the structured online HSO communication (i.e., Mode 1), and as per the findings of this study, we have formulated on a set of recommendations for (i) improving this mode of remote communication and (ii) increasing its use by the public in comparison to the other three modes. These recommendations are particularly timely given the growth in eHealth services and the lack of regulations, guidelines, and government control in relation to remote patient–physician communications. The absence of a regulatory system suggests that government bodies do not consider this a necessity, assuming perhaps that professionals should simply apply the same healthcare standards and protocols as they do in face-to-face services. However, such an approach overlooks the specific risks and opportunities associated with remote, asynchronous communications.

### 4.3. Generic Recommendations

Users vary in their capabilities and needs. For some, navigating remote communication applications could pose a challenge, especially for the older population who may be less digitally oriented. Hence, it is necessary to design a system that presents intuitive and simple-to-use interfaces and progression routes.The service is designed to providing an alternative to clinic visits (in non-urgent cases), while ensuring continuous high-quality healthcare. Hence, all requests should be fully and adequately addressed, handled by the GP, and all relevant information must be readily available to enable optimal medical decisions, including diagnoses, referrals, and treatment. Thus, the remote queries must be automatically integrated into the patient’s records, and the patient must be made aware of the GP’s response and guidelines within a reasonable period.The use of remote communication service should be restricted to patients who met with their GP in person at least once over the past year. This basic requirement ensures that the GP has the necessary contextual information for interpreting the query and formulating a suitable response.Some patients, especially younger ones who are more experienced in social media, may expect this service to be fast, immediate even; therefore, they may engage in a larger number of unfocused and imprecise communications. To mitigate the risk of placing such burden on the GPs, the number of requests per period should be limited (for example, up to four correspondences per quarter), except for emergency cases that should override this limitation.All GPs should be adequately trained on how to use this service. This is an important factor in mitigating the risk of errors associated with eHealth use. Training topics should include:
5.1.*Clinical risk mitigation*: understanding the conditions in which they can safely provide diagnoses and treatments without a physical examination.5.2.*Technology:* becoming highly familiar with the technology, including its benefits, limitations, and troubleshooting options.
The service should allow the GP to prioritize requests, receive reminders for unprocessed requests, and receive alerts when the predefined response time elapses.To increase the use of the structured online mode of communication, GPs should be instructed to discontinue any direct yet remote communications with patients (via emails or text messaging, for example—i.e., Mode 4 in this study). However, clinics as such continue to provide mediated patient–GP communications (Mode 3), as this may be the only mode of communication available for populations who are not adequately equipped to use Mode 1 (such as older adults or patients who are visually impaired). That being said, clear criteria should be developed to limit its use to such groups of patients.

### 4.4. The Content of the Remote HSO-Structured Communications Service

To ensure comprehensive, high-quality responses to the patients’ needs, the content and functionality of the service should be configured to provide a user interface that is based on the in-depth analysis of GPs’ necessary knowledge, skills, and actions.To optimize clinical decision making, including the detection of at-risk patients, the remote communication service must also incorporate the automatic and routine analysis of the content and textual features of patients’ queries, as well as patients’ demographics. This could be achieved through AI, decision-support systems, and even computerized provider order entry systems (that could send alerts in cases of contradicting medications being prescribed for a given patient).

### 4.5. The Structure and Format of the Remote HSO-Structured Communications Service

The data-gathering process should be structured (e.g., a serial format with all the relevant information-gathering fields available at each decision node) and be based on an error-reduction mechanism (e.g., preventing skipping between stages without completing logically prior fields).An option for free text, supported by a speller and with limitations on size, should be added at the end of the request to allow the patient to augment the information collected through the structured process.The instructions should be based on practical rather than abstract terminology and should be context-specific.

### 4.6. The Implementation of the Remote HSO-Structured Communications Service

When first using the service, the patient should be required to work through an online, step-by-step instruction program on how to use the service. In addition, a call center should be established to provide online support for patients using the service.Regular technology updates: the HSOs must ensure that the technology they employ is continuously updated to comply with current best clinical practices and technical stability.

### 4.7. The Sustainability of the Remote HSO-Structured Communications Service

HSOs need to continuously assess, monitor, and fine-tune the implemented service based on user feedback, especially during the initial implementation of the service, to ensure that the new technology achieves its intended outcomes. It should also be subjected to ongoing clinical review, to ensure that embedded protocols and clinical checks are in keeping with current best practices.

### 4.8. Limitations and Future Research

This novel study contributes to the literature as it sheds light on different modes of remote patient–physician communication. Although the study was carried out on modes implemented in Israel, its insights have global value as similar modes are applied worldwide.

Yet, it is important to note a number of limitations. First, the current research was designed to preliminarily identify the most desirable mode of remote patient–physician messaging for further analysis. We did not assess or claim to assess any medical outcomes associated with these different modes of communication. Future studies could benefit from conducting empirical studies for examining the clinical advantages of such remote communication modes in the healthcare service context. (Such assessments would require a costly and lengthy research program that would need to investigate numerous interacting and interrelated factors, such as HSO policies, follow-up treatments and tertiary care.)

In addition, this study focused on asynchronous remote messaging, which is just one facet of a variety of existing and emerging technologies for remote communication between GPs and patients (accounting for only 36% of all telemedicine communications [6]). Therefore, the insights gained in this study are only relevant to this specific facet and are not indicative of telemedicine in general. As such, generalizations should be made with caution. Finally, because of the uniqueness of the methodological structure of this study, it is hard to make comparisons with other similar studies.

Further studies on remote messaging should address issues that were not covered in the current study, such as the GP’s diagnostic accuracy or adherence to medical guidelines and protocols. (The ability to explore these issues is evolving thanks to the increasing use of advanced analytical tools, such as AI and big data.) Further work is also required for examining the social context, constraints, and risks that may be associated with the use of the patient’s own home as a site for initiating healthcare queries and conducting their own healthcare management. This could include an analysis of the roles of patients, caregivers, and providers as participants in the process.

## 5. Conclusions

The findings of this study indicate that patients in Israel perceive online (asynchronous) remote modes of patient–GP communications as providing near-optimal healthcare quality, since the extent to which each mode is used was found to be inversely related to its rated quality. As such, the common assumption that patients generally choose the mode of communication that is most likely to ensure high service quality is refuted. Our findings further indicate that remote services create difficulties in ensuring an accurate assessment of the patient’s clinical issue and clear communication of the recommended course of action. This is because patients often have difficulty formulating their health issues in an accurate and comprehensive manner, and/or may not always fully understand the physician’s recommendations. Governments and healthcare organizations should address these results and recommendations when developing new remote communication modes and enhancing existing ones.

## Figures and Tables

**Table 1 ijerph-20-07188-t001:** Criterion-based analysis: Performance assessment of each communication mode by criteria.

	Mode of Communication			
	1	2	3	4
Criterion	Request Submitted on HSO Website via Structured Menus	Request Submitted on HSO Website by Free Text Typing	Request Submitted by Telephoning Clinic Personnel	Request Submitted via a Free-Style Written Message to the GP’s Mobile Phone or Email
(A) Is the patient’s identity clearly specified?	The submission begins at login with a user identification process (e.g., name, birthdate, password, and ID). *Criterion is fully satisfied.*		The office representative is expected to verify the patient’s identity (name and ID). *Criterion is normally satisfied.*	The only details that appear (in relation to identity) are a phone number or an email address. Hence, the GP’s ability to identify the patient may be significantly compromised. *Criterion is rarely satisfied.*
(B) Does the submission process allow the use of clear and comprehensive specifications of the medical issue that require the GP’s attention?	The preliminary definitions of the menus and the workflow are intended to ensure the completeness and clarity of the request. However, the structured menu of questions does not always capture the patient’s situation and specific request. *Criterion is normally satisfied.*	A free text request may omit critical information due to limitations in the patient’s cognitive or linguistic skills or medical knowledge; it may also be affected by stress or discomfort. *Criterion is partially satisfied.*	The transmission of information from a patient to a staff member, who types the request, provides multiple opportunities for omitting or distorting critical information (e.g., due to communication difficulties, memory failure, or lack of understanding). *Criterion is partially satisfied.*	This mode has similar issues to those listed in mode 2 regarding typing free text. *Criterion is partially satisfied.*
(C) Is the patient’s query automatically documented in the patient’s medical records?	The request is automatically integrated into the patient’s medical records. *Criterion is fully satisfied.*	The integration of the request depends on the degree to which the GP understands the request. *Criterion is partially satisfied*	The clinic’s internal system is not linked to the patients’ records. Thus, documenting a request is at the discretion of the GP and will be prone to memory errors, distractions, omissions, etc. Also, since the system is not automated, information may be misallocated to another patient’s records. *Criterion is partially satisfied.*	The requests are not linked to the patients’ records. Thus, this mode is similar to mode 3. In addition, if the GP receives a request outside working hours, which is not unusual, the chance that the request will be entered into the medical record is further reduced. *Criterion is rarely satisfied.*
(D) Are the patient’s medical records accessible to the GP when processing the query?	Patients’ records become visible automatically upon receipt of a request, and the request’s processing is linked to the patient’s medical records. *Criterion is fully satisfied.*		The clinic’s internal system is not linked to the patients’ records. Thus, opening the records while processing the request is at the discretion of the GP. It is possible that the GP may fail to access the records or access incorrect records due to distraction or work pressure. *Criterion is partially satisfied.*	The requests are not linked to the patients’ records. Thus, this mode resembles that of mode 3. In addition, if the GP receives a request outside working hours, which is not unusual, the likelihood that the GP will consult the patient’s records is further reduced. *Criterion is rarely satisfied.*
(E) Is the GP’s answer automatically documented in the patient’s medical records?	The GP’s reply is processed and added automatically to the patient’s records. *Criterion is fully satisfied.*		The GP’s reply is documented in the patient’s records only at the discretion of the GP. Errors, omissions, and incorrect allocations may occur due to workload, distraction, or memory failure. *Criterion is partially satisfied.*	This mode has similar issues to those listed in mode 3. In addition, if the GP receives a request outside working hours, which is not unusual, the likelihood that the GP will document the reply is low. *The criterion is rarely met.*
(F) Is the maximum timeframe in which the GP must respond to a query specified in advance?	The response time for reply is predefined in the service protocol as up to five working days. In cases where the GP exceeds this period, he/she receives an alert, and the system is blocked for use until a response is provided. In cases where the GP is unavailable, the patient is automatically informed, and with his/her permission, the request is redirected to a replacement GP while maintaining the defined response time. *Criterion is fully satisfied.*		A response time is not specified as the GP (or any replacement GP) treats the request according to their availability and the perceived urgency of the request. *Criterion is not satisfied.*	This mode has similar issues to those listed in Mode 3. *Criterion is not satisfied.*
(G) Is the GP’s response clearly communicated to the patient?	The GP’s response appears on the HSO’s website. In addition, a message (SMS) is sent automatically to the patient’s mobile phone and email, confirming that a reply is available on the website. Once the patient accesses the website, he/she must confirm receipt of the message. *Criterion is fully satisfied.*		The reply is at the discretion of the GP. He/she can leave a message at his/her clinic or place an answer on the HSO’s website. *Criterion is partially satisfied.*	The reply is at the discretion of the GP. He/she can answer the patient’s email or phone directly, leave a message at his/her clinic, or place an answer on the HSO’s website. *Criterion is partially satisfied*.

**Table 2 ijerph-20-07188-t002:** Patient-perception analysis: Ratings of patients’ confidence in the communication mode’s ability to offer quality healthcare. Average score obtained for each criterion and mode of communication (range 1–4).

		Mode of Communication				
		1 (*n* = 41)	2 (*n* = 47)	3 (*n* = 36)	4 (*n* = 52)	
Statement		Request Submitted on HSO Website via Structured Menus	Request Submitted on HSO Website by Free Text Typing	Request Submitted by Telephoning Clinic Personnel	Request Submitted via a Free-Style Written Message to the GP’s Mobile Phone or Email	Mean Rating per Statement
1	The service allows me to maintain a trusting relationship with my GP	3.63	3.53	3.41	3.33	3.48
2	I can explain my request or medical problem through the remote service	3.90	3.51	3.50	3.35	3.57
3	I fully understand the medical instructions given by the GP through the remote service	3.07	3.02	3.17	3.17	3.11
4	When I request a renewal of medication or a referral to a specialist, I get an answer within a reasonable time	3.61	3.57	3.55	3.44	3.54
5	The service allows me to get adequate treatment even though I do not have to attend the clinic	3.68	3.74	3.44	3.90	3.69
6	In urgent cases, I can get an initial response within a reasonable amount of time by using the service to contact my GP	3.54	3.60	3.50	3.21	3.46
7	Communication through the service delivers proper follow-up and monitoring for illnesses, even if they are ongoing.	3.66	3.53	3.47	3.29	3.49
8	The service allows access to health services at any time and from anywhere	3.68	3.61	3.64	3.25	3.49
9	I find remote communication to be more efficient than attending the clinic	3.63	3.55	3.50	3.33	3.50
Average ± STDEV.S for each mode		3.60 ± 0.22	3.52 ± 0.20	3.47 ± 0.13	3.36 ± 0.22	
Rank order of confidence		1	2	3	4	

Note: Rank order of confidence (from 1 = high to 4 = low) according to the average scores obtained.

**Table 3 ijerph-20-07188-t003:** Ranking of the four communication modes.

	Mode of Communication			
	1	2	3	4
	Request Submitted on HSO Website via Structured Menus	Request Submitted on HSO Website by Free Text Typing	Request Submitted by Telephoning Clinic Personnel	Request Submitted via a Free-Style Written Message to the GP’s Mobile Phone or Email
Criterion-based assessment of the ability of the mode to provide quality of care	1	2	3	4
Patient-perceived confidence in meeting quality-related outcomes	1	2	3	4
Patients’ extent of use	4	3	2	1

Note: 1 = high; 4 = low.

**Table 4 ijerph-20-07188-t004:** Number (%) of queries received via each mode by patients’ age during 2016–2021.

	Mode of Communication			
	1	2	3	4
Age (Years)	Request Submitted on HSO Website via Structured Menus	Request Submitted on HSO Website by Free Text Typing	Request Submitted by Telephoning Clinic Personnel	Request Submitted via a Free-Style Written Message to the GP’s Mobile Phone or Email
<30	11,117 (29%)	8699 (23%)	4675 (12%)	13,614 (36%)
30–45	8838 (24%)	7784 (21%)	11,973 (32%)	8504 (23%)
45–60	6838 (19%)	8887 (25%)	11,244 (32%)	8372 (24%)
>60	4355 (12%)	7632 (20%)	13,638 (36%)	12,151 (32%)
SUM	31,148	33,002	41,530	42,642

The total percentage in each row = 100%.

## Data Availability

No new data were created or analyzed in this study. Data sharing is not applicable to this article.

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
