# Peer review of "Remote Communications between Patients and General Practitioners: Do Patients Choose the Most Effective Communication Routes?"

_ijerph, 2023, doi:10.3390/ijerph20247188_

Round 1

Reviewer 1 Report

Comments and Suggestions for Authors

Dear Authors,

This is an interesting study assessing the quality of remote healthcare. Below are my comments on the manuscript:

1.     Did the quality analysis take into account the security of these 4 methods of remote communication with a doctor to protect sensitive patient data?

2.     I suggest presenting more detailed data in the summary, e.g. statistical significance p

3.     Is it possible to present the sociodemographic characteristics of the patients participating in the study, or is it a homogeneous group?

4.     The number of patients in this study is low. How was the research group selected?

Author Response

Dear Reviewer,

We would like to thank you for reading our manuscript and offering insightful feedback.

Attached are two files: 1) details about changes that were made to the paper following your comments, and 2) the revised paper based on your and the other two reviewers' comments (changes are highlighted in yellow).

Yours Sincerely,

Dr. Morag (in the name of all the authors)

Reviewers' comments

Answers / Changes made in the manuscript

Reviewer 1

1.   Did the quality analysis take into account the security of these 4 methods of remote communication with a doctor to protect sensitive patient data?

In regard to the first analysis, 'Criterion-based analysis', the following is stated in the main text (p. 4, line 197) "It is important to note that due to the difficulty in interpreting messages without knowledge of the given medical context, the specific communication messages between the GPs and their patients were not analyzed"; meaning, the actual medical data was not available for the researchers. As for the second analysis, 'Patients’ perceptions of healthcare quality', the patients were asked about their experience of interacting with one of the four remote communication modes without any reference to their medical status or needs.

2.   I suggest presenting more detailed data in the summary, e.g. statistical significance p

All the statistical analyses that were carried out in the study are detailed in the Results section, including probabilities (See p.10, line 310; p.12, line 359; and p.13, line 361). These are not repeated in the Discussion section, to reduce confusion and redundancy.

3.    Is it possible to present the sociodemographic characteristics of the patients participating in the study, or is it a homogeneous group?

In the first two analyses, 'Criterion-based analysis' and 'Patients’ perceptions of healthcare quality,' the patients' sociodemographic characteristics (e.g., gender, residence, or age) were not disclosed to the researchers, as they were irrelevant to the aim of the evaluations.

In the third analysis, we were only interested in the patients' age, to ensure that they are not legally considered minors.

We have, however, added a sentence to the Method, section 2.3 (p. 6, line 262), that clarifying the minimum age for participation in the survey.

4.   The number of patients in this study is low. How was the research group selected?

As we were aware that the response rate (44%) may pose a methodological “pothole,” we conducted a-prior power analysis using G*Power3, while applying a one-way ANOVA test and an alpha level of 0.05. This analysis yielded a power of 89.6%, which is considered very high compared to the literature (please see p.10, line 307).   

Reviewer 2 Report

Comments and Suggestions for Authors

The study is very innovative and shows interesting results about remote communication between GPs and patients. It is a unique study, which therefore lacks similar options with the cauels to compare. This is a limitation in the discussion, however, it is achieved in a way that captures the reader's interest. It is a topic that is highly relevant for the planning of current and future clinical management platforms.

​The methods do not mention the total universe of users of the health system nor do they calculate a minimum sample number for each mode of communication. It was decided to send 100 surveys per mode of communication, whose response rate was low and we do not know if it ensures the minimum sample number to have statistical power. For example, mode 3 has only 36 responses.

This is a limitation of the study, which is not mentioned and which may be a methodological obstacle.There are formatting details to improve:

In results, table 4 is presented with the communication modalities by age groups in years prior to the study. I suggest incorporating it into the introduction as an input of context for the study. Another option is to include it in the method and report it as the first results.

Table 2 presents the confidence range values (1=high to 4=low) in its last row. This row should be placed as a footnote to the table. It is not defined what the values presented correspond to. I can assume that it is the average of the score obtained, but it is not clear to me. It must be specified in the table title: average score obtained for each criterion and mode of communication (range 1-4).

Point 3.3 mentions "correlation", however, the results in table 3 do not present correlation analysis. I suggest changing the word, as it leads to confusion. In the same table the bottom line of the second row is missing. The format of the lines is not uniform.

Finally, the conclusions must answer the main question or aim of the study. I suggest copying and improving what is described in the summary between lines 17 to 27.

Author Response

Dear Reviewer,

We would like to thank you for reading our manuscript and offering insightful feedback.

Attached are two files: 1) details about changes that were made to the paper following your comments, and 2) the revised paper based on your and the other two reviewers' comments (changes are highlighted in yellow).

Yours Sincerely,

Dr. Morag (in the name of all the authors)

Reviewers' comments

Answers / Changes made in the manuscript

Reviewer 2

1.   It is a unique study, which therefore lacks similar options with the cauels to compare. This is a limitation in the discussion

Thank you for this important comment. The following sentence has now been added to Section 4.8 (Limitations and future research; p.18, line 601):

“Finally, because of the uniqueness of the methodological structure of this study, it is hard to make comparisons with other similar studies.”

2.   The methods do not mention the total universe of users of the health system nor do they calculate a minimum sample number for each mode of communication. It was decided to send 100 surveys per mode of communication, whose response rate was low and we do not know if it ensures the minimum sample number to have statistical power. For example, mode 3 has only 36 responses.

This is a limitation of the study, which is not mentioned and which may be a methodological obstacle.

As we were aware that the response rate (44%) may pose a methodological “pothole,” we conducted a-prior power analysis using G*Power3, while applying a one-way ANOVA test and an alpha level of 0.05. This analysis yielded a power of 89.6%, which is considered very high compared to the literature (please see p.10, line 307). 

3.   In results, table 4 is presented with the communication modalities by age groups in years prior to the study. I suggest incorporating it into the introduction as an input of context for the study. Another option is to include it in the method and report it as the first results.

Thank you for this comment. This limitation has now been incorporated into Section 2.4: “The use of each mode in practice,” p.6, line 277.

4.   Table 2 presents the confidence range values (1=high to 4=low) in its last row. This row should be placed as a footnote to the table. It is not defined what the values presented correspond to. I can assume that it is the average of the score obtained, but it is not clear to me. It must be specified in the table title: average score obtained for each criterion and mode of communication (range 1-4).

These suggested clarifications have been added to Table 2, in both the title and as a footnote (please see p. 11).

5.  Point 3.3 mentions "correlation", however, the results in table 3 do not present correlation analysis. I suggest changing the word, as it leads to confusion. In the same table the bottom line of the second row is missing. The format of the lines is not uniform.

The word "correlation" has been changed to "interrelationships" (please see p. 12, line 328). In addition, the formatting of the lines in Table 3 has been corrected.

6.   Finally, the conclusions must answer the main question or aim of the study. I suggest copying and improving what is described in the summary between lines 17 to 27.

Thank you for this important comment. The Conclusions section has been updated to include answers to the research question and reflect the study's insights. (Please see p.18, line 619.)

Reviewer 3 Report

Comments and Suggestions for Authors

Thank you for the opportunity to review this paper. I found it very interesting, informative and well-written. I have only few comments for the authors on aspects that I think that could be improved:

-        It would seem a good idea to explain in more detail why these 2 of 9 criteria “do not contribute or add value to the quality of the healthcare provided.” – what were these? why were they identified at all if they were not used later? it is mainly unclear since we do not know what they were

-        I noticed that in the Results section authors sometimes tend to explain their procedure and these methods fragments are mixed with the actual results decreasing the readability. I would suggest explaining the procedure in detail in the Methods section (next to the paragraph on statistical analysis) and in the Results focus only on results.

-        there seems to be a formatting issue between lines 465-468 (5.1 and 5.2)

-        some of the most important (in authors’ opinion) recommendations provided in the discussion could also be briefly repeated/summarized in the conclusion section – I think this could improve the discoverability of the study

-        after reading the whole manuscript, I would also suggest modifying the abstract to more precisely indicate its usual fragments (introduction/aim/methods/results/conclusion) – they do not have to be named but their fragments should be clearly included (just as in the Journals’ template guidelines). For me, now especially the methods part is not clearly described. Also the results/conclusion part could be clearer and less intertwined – the sole fact that the authors feel that they have to explain previous sentence should be an indicator of that (“In other words, …” – line 19). I suggest that the authors should simply 1) summarize their main findings and then 2) briefly discuss them and conclude the paper.

Author Response

Dear Reviewer,

We would like to thank you for reading our manuscript and offering insightful feedback.

Attached are two files: 1) details about changes that were made to the paper following your comments, and 2) the revised paper based on your and the other two reviewers' comments (changes are highlighted in yellow).

Yours Sincerely,

Dr. Morag (in the name of all the authors)

Reviewers' comments

Answers / Changes made in the manuscript

Reviewer 3

1.  It would seem a good idea to explain in more detail why these 2 of 9 criteria “do not contribute or add value to the quality of the healthcare provided.” – what were these? why were they identified at all if they were not used later? it is mainly unclear since we do not know what they were

Clarifications regarding the reason for the omission of these two factors has been added to Section 2.2 (Please see p.5, line 204).

2.   I noticed that in the Results section authors sometimes tend to explain their procedure and these methods fragments are mixed with the actual results decreasing the readability. I would suggest explaining the procedure in detail in the Methods section (next to the paragraph on statistical analysis) and in the Results focus only on results.

Could please you give an example of this, so that we can address the issue that you are referring to?

3.   There seems to be a formatting issue between lines 465-468 (5.1 and 5.2)

The file has been processed by the journal’s Editorial office, and currently we do not see such a problem. Sometimes, Word files change after being sent by email to other readers.

4.   Some of the most important (in authors’ opinion) recommendations provided in the discussion could also be briefly repeated/summarized in the conclusion section – I think this could improve the discoverability of the study

Thank you for this important comment. The Conclusions section has been updated to include answers to the research question and reflect the study's insights. Please see p.18, line 619.

5.   After reading the whole manuscript, I would also suggest modifying the abstract to more precisely indicate its usual fragments (introduction/aim/methods/results/conclusion) – they do not have to be named but their fragments should be clearly included (just as in the Journals’ template guidelines). For me, now especially the methods part is not clearly described.

Also the results/conclusion part could be clearer and less intertwined – the sole fact that the authors feel that they have to explain previous sentence should be an indicator of that (“In other words, …” – line 19). I suggest that the authors should simply 1) summarize their main findings and then 2) briefly discuss them and conclude the paper.

Thank you. We have rewritten the Abstract to ensure clarity.

Round 2

Reviewer 2 Report

Comments and Suggestions for Authors

The authors have considered all the commnets and suggestions I made.